# A molecular movie of ultrafast singlet fission

Christoph Schnedermann [1,2], Antonios M. Alvertis [1], Torsten Wende[2], Steven Lukman[1,8], Jiaqi Feng[3], Florian A.Y.N. Schröder[1], David H.P. Turban[1], Jishan Wu [3], Nicholas D.M. Hine[4], Neil C. Greenham[1], Alex W. Chin[5], Akshay Rao [1], Philipp Kukura [2] & Andrew J. Musser [6,7]

The complex dynamics of ultrafast photoinduced reactions are governed by their evolution along vibronically coupled potential energy surfaces. It is now often possible to identify such processes, but a detailed depiction of the crucial nuclear degrees of freedom involved typically remains elusive. Here, combining excited-state time-domain Raman spectroscopy and tree-tensor network state simulations, we construct the full 108-atom molecular movie of ultrafast singlet fission in a pentacene dimer, explicitly treating 252 vibrational modes on 5 electronic states. We assign the tuning and coupling modes, quantifying their relative intensities and contributions, and demonstrate how these modes coherently synchronise to drive the reaction. Our combined experimental and theoretical approach reveals the atomic-scale singlet fission mechanism and can be generalized to other ultrafast photoinduced reactions in complex systems. This will enable mechanistic insight on a detailed structural level, with the ultimate aim to rationally design molecules to maximise the efficiency of photoinduced reactions.

[1] Cavendish Laboratory, University of Cambridge, JJ Thomson Avenue, Cambridge CB3 0HE, UK. [2] Physical and Theoretical Chemistry Laboratory, Oxford University, South Parks Road, Oxford OX1 3QZ, UK. [3] Department of Chemistry, National University of Singapore, 3 Science Drive 3, Singapore 117543, Singapore. [4] Department of Physics, University of Warwick, Gibbet Hill Road, Coventry CV4 7AL, UK. [5] Centre National de la Recherce Scientifique, Institute des Nanosciences de Paris, Sorbonne Universite, Paris, France. [6] Department of Physics and Astronomy, University of Sheffield, Hounsfield Road, Sheffield S3 7RH, UK. [7] Department of Chemistry and Chemical Biology, Cornell University, Baker Laboratory, Ithaca, NY 14853, USA. [8] Present address: Institute of Materials Research and Engineering, Agency for Science Technology and Research (A*STAR), 2 Fusionopolis Way, Singapore 138634, Singapore. Correspondence and requests for materials should be addressed to C.S. (email: cs2002@cam.ac.uk) or to A.J.M. (email: ajm557@cornell.edu)

The coupling between vibrational and electronic degrees of freedom after photon absorption defines photochemical reaction pathways and guides processes such as charge and exciton generation[1,2], transport[3,4] and recombination[5] as well as photoisomerisation[6] and bond-dissociation[7]. The initial photo-reactivity after photoexcitation is governed by ultrafast processes, including a correlated evolution along vibrational coordinates and their associated electronic states on the reaction coordinate. As a result, these processes cannot be described in the framework of the Born-Oppenheimer approximation[1,8–10]. Despite remarkable progress in the optical manipulation of vibrational and electronic states[3,5] and the identification of vibronically coherent processes[1,8,11,12], the precise molecular mechanisms and associated structural changes remain largely elusive and subject to competing interpretations.

This uncertainty stems from a disparity between experimental and theoretical methods. Structurally sensitive experimental techniques that can access the earliest photoreactive transformations are often only available for large and complex molecular systems, while accurate first-principles computational modelling for such non-Born-Oppenheimer dynamic processes is only affordable for much smaller models[13,14]. Consequently, even if experimental structural information is available, it can rarely be accurately projected onto the molecular origin of the crucial coupling and tuning modes involved, preventing identification of the operative reaction mechanisms[15,16]. Developing a detailed molecular understanding of such complex photoinduced processes is crucial, however, to provide rational design criteria for improved functional materials, for instance for organic optoelectronics and molecular photocatalysts.

To this end, novel theoretical methods that can address complex molecular systems are critical. The experimental validation of such theories must go beyond simple population dynamics—too coarse a figure of merit—and explicitly include structural configuration changes after photoexcitation[17]. In this study, we demonstrate the power of such a combined approach as applied to the model process of singlet fission, i.e. the conversion of a photoexcited singlet exciton ($S_1$) into two triplet excitons via a correlated triplet pair ($^1TT$) intermediate[18].

Singlet fission is a classic example of an ultrafast process in which the molecular mechanism can only be inferred due to the lack of experimentally and theoretically comparable data sets, despite extensive study over the past decade[19–23]. In recent years, it has been demonstrated in several thin-film singlet fission systems that the initial $S_1$-$^1TT$ conversion is vibrationally coherent[11,12,24–26]. Other studies using structurally sensitive techniques have also found that the key electronic processes in singlet fission are linked to inter- and intramolecular motions[27,28]. However, the structural complexity of these systems precluded direct interpretation in terms of specific motions and their role in the reaction. Within the theoretical community, studies have shown that the typical vibronic couplings in singlet fission materials are strong (10's to 100's meV) and thus require non-perturbative methods to be accurately described[25,29,30], and the same is true in many other molecular systems[31]. As a result, advanced simulation techniques have been applied to elucidate the varied roles of ultrafast and non-equilibrium environmental dynamics on fission[32–35]. There is growing recognition that singlet fission, like the majority of ultrafast (<10 ps) processes, is intimately coupled to nuclear dynamics[23,29,30,36–42]. Nonetheless there is no clear determination of what motions drive the process, how this coupling occurs, or indeed whether the reported vibrational coherence is important in achieving a high reaction yield or simply a consequence of the ultrafast nature of the reaction with no functional importance. These problems are typical of the more general study of non-Born-Oppenheimer

dynamics, and they constitute a key bottleneck in materials understanding and design.

One example system in which such ultrafast structural changes are expected to play a significant role in the electronic dynamics is the large (108-atom) and complex pentacene dimer DP-Mes (Fig. 1a). Like many dimers of pentacene[43–45], DP-Mes is capable of sub-ps intramolecular singlet fission[46,47]. The reaction rates and $^1TT$ yields depend strongly on solvent environment[46,47], indicating that fission is mediated by coupling to higher-energy charge-transfer states[22,38,39,42] (Fig. 1b). We have previously simulated the full structural dynamics of singlet fission in DP-Mes using a recently developed, fully quantum-mechanical Tree Tensor Network state (TTNS) algorithm[48]. The TTNS method is largely based on the formalism of matrix product states and tree tensor states, which have been successfully applied to condensed matter problems[49–54] and—more recently—to the dynamics of open quantum systems[55–61]. In our previous work on DP-Mes[48], a combination of machine learning and entanglement renormalisation techniques was employed to capture the non-perturbative and non-Markovian physics arising from strong coupling to a large number of vibrational modes (see Methods and Supplementary Discussion, section 1). A recent theoretical study highlighted the breakdown of perturbative approaches in predicting singlet fission rates in pentacene dimer systems[62], further emphasising that non-perturbative approaches such as the TTNS method[48] are necessary to describe fission dynamics. Our earlier DP-Mes simulation confirmed that singlet fission is mediated by super-exchange and is driven by a chain of cooperative vibronic processes involving modes of different timescales and symmetries.

In this work, we employ ultrafast excited-state vibrational spectroscopy to probe the transfer of vibrational wavepackets from the initially photoexcited $S_1$ state to the $^1TT$ state in DP-Mes. We build on the previous TTNS simulations to map the structural dynamics to real experimental observables. This exact quantum treatment is uniquely suited to describe systems such as DP-Mes, which have strong vibronic couplings (up 0.3 eV, Supplementary Tables 2–3). The model considers five excited electronic states—two symmetry-adapted singlet states, two charge-transfer states and $^1TT$—and 252 strongly coupled vibrational modes whose properties are obtained from ab initio electronic structure techniques[47]. The simulations closely reproduce both the frequencies and the intensities of the experimentally retrieved vibrational coherence signatures. The remarkable structural agreement between theory and experiment enables us to reconstruct the real-space structural motion associated with the vibronic dynamics as a real-time movie, giving a fully quantum visualisation of the process and enabling determination of the nature of the critical coupling and tuning modes driving this coherent ultrafast reaction. Our results provide a refined atomistic picture of the molecular mechanism of singlet fission and allow us to visualise ultrafast quantum dynamics in an experimentally verifiable way.

## Results

**Vibrationally coherent singlet fission.** We use femtosecond transient absorption spectroscopy with a time resolution of ~13 fs to track the singlet fission process in DP-Mes (Fig. 1c). The strong positive bands ($\Delta T/T > 0$) match the ground-state absorption peaks (grey spectrum) and can be attributed to a ground-state bleach signal. At early time delays (<0.7 ps) this bleach is spectrally overlaid with the characteristic stimulated emission ($\Delta T/T > 0$) of $S_1$ in the range 625–725 nm, in agreement with the very short-lived[46] photoluminescence (orange spectrum). The $S_1$ state decays concurrently with the rise of a distinctive excited-state absorption ($\Delta T/T < 0$)

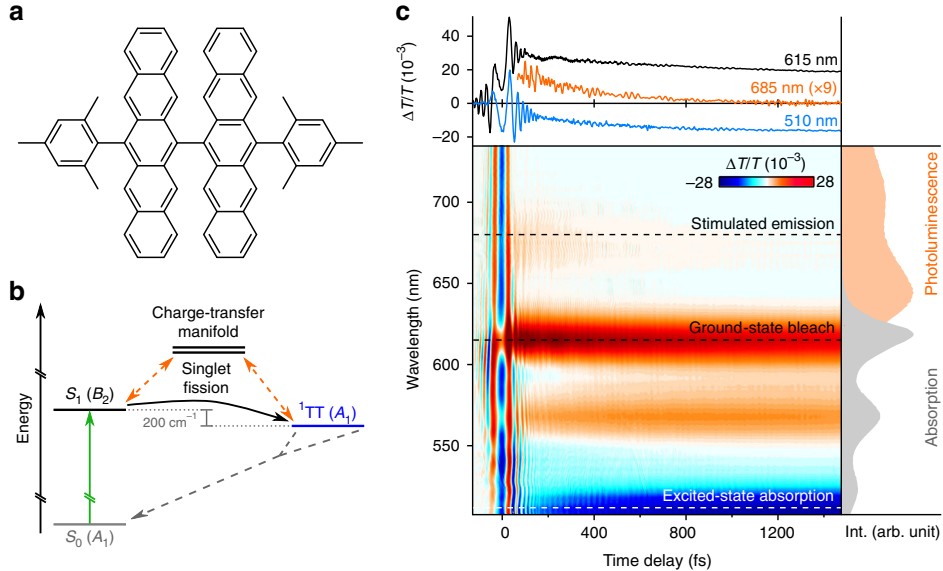

**Fig. 1** Structural and optical properties of DP-Mes. **a** Chemical structure of DP-Mes. Orthogonal ground-state geometry shown in Supplementary Fig. 1. **b** DP-Mes electronic states and photophysics. Singlet fission from $S_1$ to $^1TT$ is mediated by coupling (orange arrows) to a manifold of charge-transfer states which are not directly populated, facilitating the overall marginally exothermic (~200 $cm^{-1}$) process[46, 47]. Symmetries of the relevant excited electronic states are indicated in parentheses[48]. **c** Differential transmission map of a DP-Mes thin film following excitation with a 13 fs pulse centred at 550 nm, at room temperature. The absorption (grey) and photoluminescence (orange) spectra are shown to guide spectral assignment. The primary spectral features of $S_1$ (stimulated emission), $^1TT$ (excited-state absorption) and total excited-state population (ground-state bleach) are indicated by dashed lines. The associated transient kinetics are shown above, with the weak stimulated emission trace scaled by a factor of 9 for clarity. The superimposed oscillatory modulations correspond to vibrationally coherent wavepackets formed by impulsive excitation. The transient absorption spectrum over an extended probe wavelength range is presented in Supplementary Fig. 9

peaked at ~515 nm previously assigned to $^1TT$[46,47], with a time constant of 320 fs. These dynamics are consistent with previous reports using low-power narrow-band excitation and reveal highly efficient (>90%) singlet fission[46,47].

The electronic population dynamics show distinct oscillatory modulations throughout the probed spectral window, which report on impulsively generated vibrational wavepacket motion on both ground- and excited-state potential energy surfaces[63–66]. This vibrational coherence can be isolated through subtraction of the slower electronic dynamics at each detection wavelength followed by Fourier transformation, which yields the impulsive Raman spectrum detected at each probe wavelength (Fig. 2a).

Three distinct spectral regions of vibrational coherence activity are discernible in Fig. 2a, matching the ground-state bleach, stimulated emission and excited-state absorption features identified above (Fig. 1c). The impulsive Raman spectrum in the ground-state bleach region (Fig. 2b, black) exhibits several peaks in the high-frequency region at 1153, 1210 and 1372 $cm^{-1}$ as well as an intense low-frequency mode at 263 $cm^{-1}$ and a weaker mode at 785 $cm^{-1}$. The Raman spectrum is in good agreement with a ground-state impulsive Raman spectrum measured separately as a reference (Fig. 2b, grey spectrum, experimental details in Methods), indicating that this spectral region is dominated by vibrational activity on the ground state, $S_0$. We attribute the difference in relative peak intensities between the two $S_0$ spectra, especially in the low-frequency region, to the different pump resonance enhancement conditions employed[67].

The impulsive Raman spectrum obtained in the stimulated emission region (Fig. 2b, red) reveals similar peak positions to the ground-state spectrum at 263, 608, 785, 1160, 1198 and 1372 $cm^{-1}$. These are slightly shifted (<10 $cm^{-1}$) and exhibit markedly different intensity profile, especially in the high-frequency region. Based on these subtle differences, we tentatively assign this

spectrum to the excited $S_1$ state, albeit with underlying ground-state contributions preventing unambiguous assignment (see Supplementary Discussion, section 2)[68].

In the excited-state absorption band (Fig. 2b, blue) the impulsive Raman spectrum reveals pronounced differences from the ground-state region both in relative peak intensities and frequencies, allowing confident assignment to the $^1TT$ state. Compared with the $S_0$ and $S_1$ Raman spectra (Fig. 2b, black and red), the $^1TT$ Raman spectrum exhibits higher-frequency peak positions at 793, 1335 and 1392 $cm^{-1}$ as well as novel bands at 127 and 1126 $cm^{-1}$. The observation of distinct vibrational coherence in $^1TT$, which is not directly photoexcited, has important mechanistic consequences. It demonstrates that the molecular vibrations initiated on photoexcitation are precisely synchronised with the change in electronic state and accompanying shifts in frequency, ruling out stochastic hopping or tunnelling between $S_1$ and $^1TT$ surfaces. Such a phenomenon could be explained through smooth evolution along a simple adiabatic potential energy surface from $S_1$-rich to $^1TT$-rich character[21], but this is not the case in DP-Mes: $S_1$ and $^1TT$ exhibit negligible mixing at the orthogonal Franck-Condon point[46,47] and our recent TTNS simulations indicate the fission process is driven by non-adiabatic coupling through vibronic super-exchange[48]. Instead, the finding of vibrational coherence across the full vibrational fingerprint region in $^1TT$ requires a vibrationally coherent process in which the photogenerated vibrational wavepackets on $S_1$ are transferred to $^1TT$[8]. Earlier studies of intermolecular singlet fission in TIPS-pentacene and other acene films reported a similar behaviour[11,24,25]. The intramolecular process in DP-Mes thus follows the same mechanism identified for intermolecular fission in TIPS-pentacene, even though it proceeds substantially slower in the dimer system (320 fs vs 80 fs).

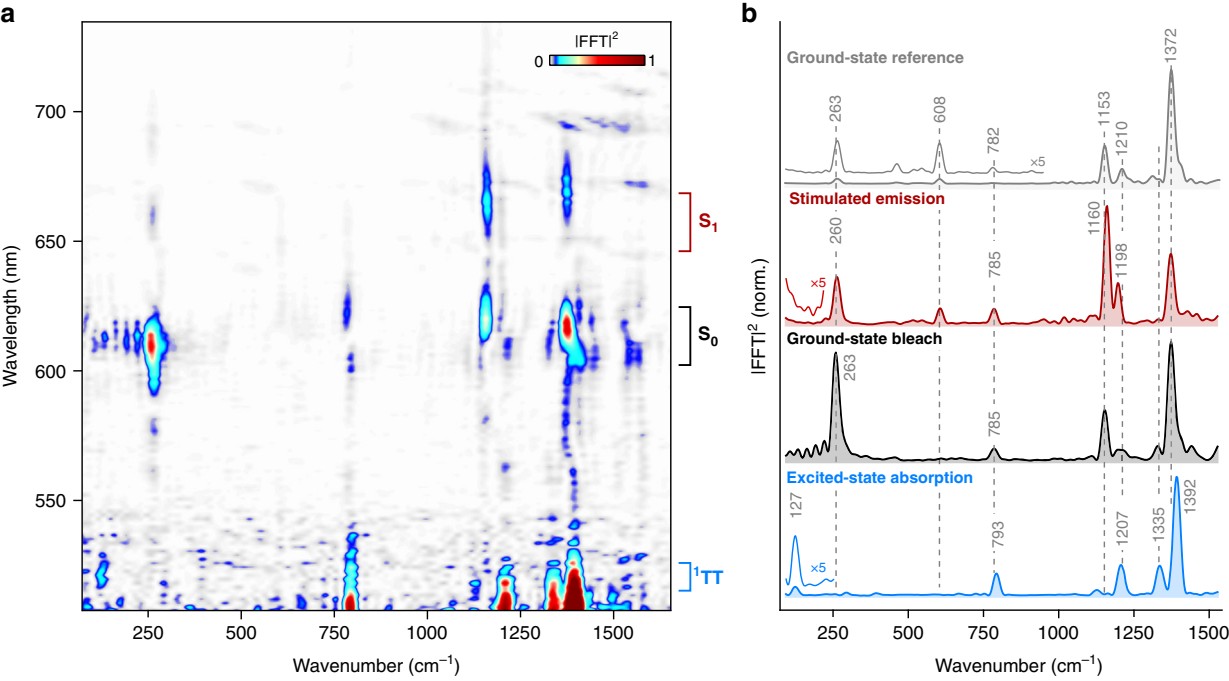

**Fig. 2** Impulsive vibrational spectroscopy of DP-Mes. **a** Wavelength-resolved impulsive Raman map of DP-Mes thin film following impulsive photoexcitation into $S_1$. FFT: fast Fourier transform. **b** Impulsive Raman spectra for different spectral components, integrated as shown by brackets in (**a**): stimulated emission —645–665 nm (red, $S_1$), ground-state bleach—605–625 nm (black, $S_0$), excited-state absorption—515–525 nm (blue, $^1$TT). The off-resonant impulsive Raman spectrum of DP-Mes in $S_0$ averaged over probe wavelengths from 605 to 625 nm is shown for comparison (grey, details in Methods). Dashed vertical lines highlight the difference between ground-state vibrational modes and those identified in $^1$TT. All spectra are normalised to the strongest peak

**Identification of transferred coherence**. To elucidate the origin of the different contributions in the $^1$TT vibrational coherence spectrum and how it is affected by the $S_1 \rightarrow {}^1$TT crossing event, we carried out impulsive Raman reference measurements on $^1$TT after the initial singlet fission dynamics were complete (9 ps after photoexcitation, with a Raman pulse tuned to the $^1$TT excited-state absorption, see Methods for details)[66]. In Fig. 3, we compare the resulting intrinsic $^1$TT Raman spectrum (purple) with the transferred $S_1 \rightarrow {}^1$TT spectrum isolated earlier (blue). Both exhibit the same high-frequency signature bands (>1200 cm$^{-1}$) with similar intensity ratios. In the lower-frequency region, however, we observe numerous modes (127, 793, 1126 and 1207 cm$^{-1}$) with strongly enhanced intensities in the $S_1 \rightarrow {}^1$TT Raman spectrum.

Studies of photoinduced internal conversion in β-carotene[69] and rhodopsin[70] suggest that vibrational modes which show similar frequencies and relative intensities in both Raman spectra (intrinsic and transferred) can be assigned to tuning modes of the underlying photochemical process. Such modes are required in a process mediated by a conical intersection or avoided crossing to yield electronic degeneracy between initial and final electronic states, but they take no active part in the reaction[15]. In contrast, the presence of additional modes in the transferred coherence Raman spectrum has been largely unexplored. While the exact nature of these modes cannot be obtained from our experiments, the difference between transferred and intrinsically generated impulsive Raman spectra reflects a fundamental difference in the mechanism of vibrational coherence generation (see Supplementary Discussion, section 3, for further discussion).

**Simulation of full quantum dynamics**. To gain structural insight into this vibrationally coherent mechanism, we modelled the full quantum dynamics of fission employing a recently developed TTNS approach[48] which accounts for 252 vibrational modes

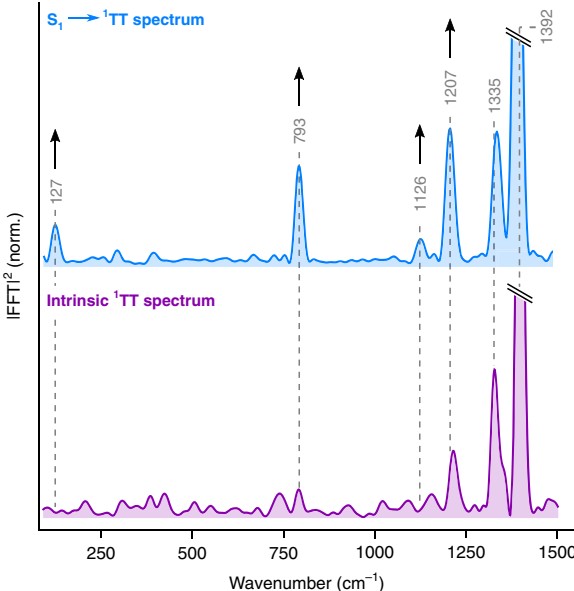

**Fig. 3** Identification of transferred coherence. Comparison of impulsive Raman spectra directly generated in $^1$TT (bottom, purple) and transferred from $S_1$ via singlet fission (top, blue, reproduced from Fig. 2). Experimental spectra are normalised to the high-frequency peak at 1392 cm$^{-1}$, which is truncated for clarity. Arrows indicate regions where singlet fission increases the peak (coherence) intensity. Dashed vertical lines are a guide to the eye indicating the dominant vibrational modes

spanning 110–1680 cm$^{-1}$ and their respective couplings to 5 excited electronic states (see Methods). This method expands the full vibronic wavefunction of the system into a network of tensors which represent molecular vibrations of different symmetries as well as the electronic system (Supplementary Figs. 3–5). All

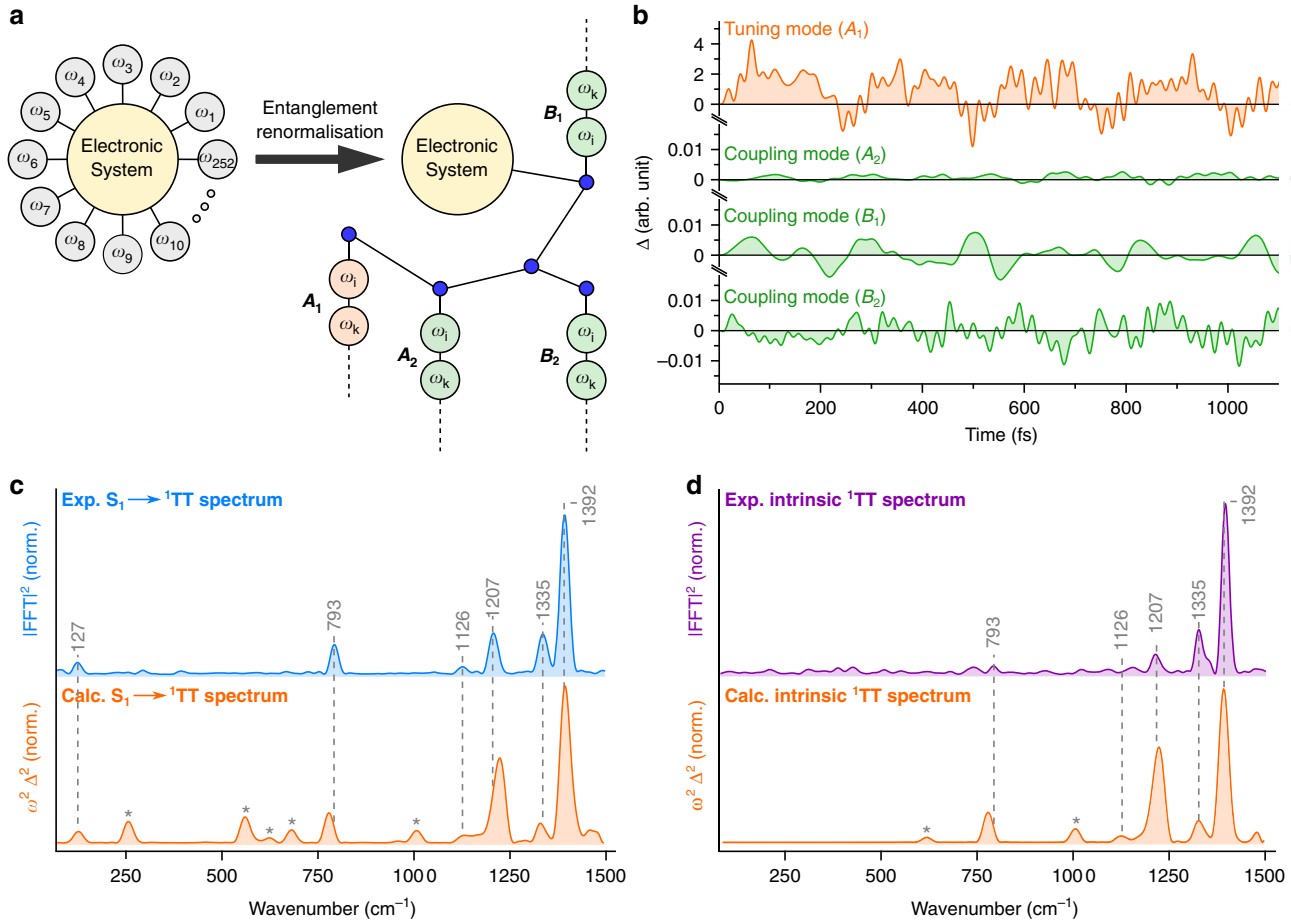

**Fig. 4** TTNS simulation of singlet fission in DP-Mes and structural experimental comparison. **a** Schematic of TTNS. The electronic system is coupled to 252 vibrational modes ($\omega$) in DP-Mes. By means of entanglement renormalisation, a tree of linearly connected vibrational modes connected to the electronic system via entanglement renormalisation nodes (blue) can be created, allowing facile computation of non-Born-Oppenheimer dynamics for large molecules (>100 atoms, further details in Methods). The vibrational modes are grouped by symmetry and colour coded to indicate tuning (orange) or coupling (green) mode behaviour. **b** Projected total time-dependent mode displacements for all symmetry groups and their corresponding assignment. We remark the ~200-fold lower displacement amplitude for $A_2$, $B_1$ and $B_2$ modes compared with the $A_1$ modes. **c** $S_1 \rightarrow {}^1TT$ coherence transfer Raman spectrum (blue) and resonance Raman representation of the calculated spectrum (orange). **d** Intrinsic ${}^1TT$ Raman spectrum (purple) compared with the resonance Raman representation of calculations initiated in ${}^1TT$ (orange). Calculated modes marked with an asterisk in (**c**) and (**d**) are not detected in the experimental spectrum, likely due to resonance Raman enhancement effects

groups of vibrational tensors are connected (i.e. coupled) to the tensor representing the electronic system (Fig. 4a, left). This wavefunction is evolved in time using the time-dependent variational principle[71] in combination with a linear vibronic Hamiltonian, parametrised from ab initio calculations of the electronic and vibrational structure of DP-Mes (Methods and Supplementary Discussion, section 1). Crucially, not all correlations between the elements of the system are required to correctly capture its underlying ultrafast dynamics[48]. Upon re-expanding the wavefunction into a tree structure by means of entanglement renormalisation (Fig. 4a, right, and Supplementary Figs. 4, 5), we can include only the most significant correlations, making calculation of the time evolution computationally feasible even for large systems such as DP-Mes[51,72]. This approach allows numerically exact quantum-mechanical treatment of many-body vibronic wavefunctions, without any recourse to perturbation theory or loss of information, as in reduced density matrix approaches that invoke Markov-like approximations. We note that despite the fact that not all of the 252 modes are strongly excited over the course of the dynamics, it remains important to include them in order to capture the effects of dissipation and

irreversibility that a large vibrational bath imposes. This capability is of paramount importance for modelling ultrafast molecular photophysics—as well as a wide range of other nanoscale systems—as the concurrent time evolution of both the electronic and vibrational degrees of freedom is an essential part of the non-Born-Oppenheimer and emergent entanglement dynamics in such systems.

Though the simulations are performed at absolute zero, this theoretical model correctly reproduces the essential room temperature electronic photophysics of DP-Mes—quantitative singlet fission from $S_1$ to ${}^1TT$, mediated by coupling to charge-transfer states which are not directly populated—and provides full dynamic information about the nuclear motion[48] (see Supplementary Fig. 11 for a brief discussion of temperature effects). Based on the simulations we can examine the role of each vibrational mode during the singlet fission reaction in DP-Mes and its contribution to the experimentally observed impulsive Raman spectrum. To accurately compare the calculations to the experimental results it is important to note that the recorded impulsive Raman intensity depends on the dimensionless displacement parameter, $\Delta$, of the state-specific vibrational

normal modes of DP-Mes according to Eq. (1), where $\omega$ denotes vibrational frequency.

$$I \propto \Delta^2 \omega^2 \qquad (1)$$

We therefore expanded the TTNS approach by projecting the calculated time-dependent displacements of the tensor states onto the molecular normal-mode vibrations of DP-Mes determined by density-functional theory (see Supplementary Discussion, section 1.3). It is instructive to discuss these time-dependent normal-mode displacements according to their symmetry properties by computing the total displacement amplitude for each symmetry group (Fig. 4b). Intriguingly, the most active modes belong to $A_1$ symmetry (Fig. 4b, top) while the remaining modes ($A_2$, $B_1$ and $B_2$) are less displaced by at least a factor of 200. The form of the calculated coupling matrices (see Supplementary Discussion, section 1.1), furthermore allows us to characterise modes of $A_1$ symmetry as tuning modes (Fig. 4b, orange), while the modes of $A_2$, $B_1$ and $B_2$ symmetry act as coupling modes of the singlet fission reaction (Fig. 4b, green). This behaviour is expected for the vibronically coupled $S_1(B_2) \rightarrow {}^1TT(A_1)$ singlet fission reaction, as this process requires some form of symmetry-breaking motion to occur (see Fig. 1b)[48].

**Benchmarking against vibronic spectroscopy.** To directly compare the theoretical results to the experimentally obtained coherence transfer Raman spectrum, we computed the Fourier transform of the total time-dependent normal-mode displacements and converted the retrieved displacement amplitudes ($\Delta$) into resonance Raman spectra from 100 to 1500 cm$^{-1}$ according Eq. (1) (Fig. 4c, orange, see Supplementary Discussion, section 1.4, for details). We observe remarkable agreement with the coherence transfer Raman spectrum (Fig. 4c, blue), with clear activity not only in the prominent 1335 and 1392 cm$^{-1}$ modes but also reproducing the spectrum at 127, 793, 1126 and 1207 cm$^{-1}$ with a frequency accuracy of <20 cm$^{-1}$ and an excellent match to the intensity profile throughout the entire frequency region.

Our calculation also predicts several other strongly displaced modes in $^1TT$, in particular in the region 250–800 cm$^{-1}$, which are not observed experimentally (asterisks in Fig. 4c). Whereas the simulation describes the full Raman spectrum of vibrational coherence generated after photoexcitation and transferred through singlet fission, we can only probe this coherence via the excited-state absorption transition at ~520 nm. If a vibrational mode is not displaced along this (probed) transition, our experiment cannot benefit from the associated resonance Raman enhancement and will not display significant Raman intensity. Such a mode-specific resonance Raman effect is also commonly observed in linear resonance Raman spectroscopy upon tuning the excitation wavelength into different electronic transitions[67]. We therefore believe that the modes absent in the experimental spectrum in the region from 250 to 800 cm$^{-1}$ are not Raman active on the $T_1 \rightarrow T_3$ absorption transition and are consequently invisible to our experiment. We expect that exact modelling of the resonance-specific Franck-Condon factors, as reported for TIPS-pentacene[73], would improve the match between experiment and theory, but such calculations are extremely demanding and beyond the scope of this work. Instead, we emphasize the overall intensity agreements, particularly in the high-frequency region (>1000 cm$^{-1}$). Such agreement, despite the fact our resonance Raman representation of the calculated displacements is only an approximation, supports the notion that the observed intensity differences between experiment and theory are related to mode-specific resonance Raman effects.

Two other potential explanations for these differences can be considered. Firstly, it has previously been reported that the singlet

fission dynamics of DP-Mes are highly dependent on the environment[46,47], a parameter not incorporated in the simulations. We anticipate that any change in the underlying mechanism will further affect the relative intensities of the observed Raman modes, complicating absolute intensity comparison between experiment (thin film) and theory (vacuum). It is encouraging in this regard that the experimental spectrum does not contain modes which were not predicted by the simulation. Secondly, we can consider that the additional modes (Fig. 4, asterisks) in the simulation arise coincidentally from the calculation. Since these modes cannot be detected in our experiment, we cannot directly confirm that they correspond to real molecular displacements. However, examining the absolute displacement parameters, we find that the low-frequency modes are among the most displaced in the entire system and therefore largely responsible for the overall reaction dynamics (Supplementary Figs. 6, 7). Consequently, we would not expect our model to nearly quantitatively reproduce the timescale of singlet fission, the transfer of vibrational coherence and even the sensitivity of the $^1TT$ vibrational coherence to the way it is generated (see below), if these modes were spurious in origin. This accuracy, benchmarked on multiple observables, suggests that the underlying linear vibronic Hamiltonian provides a satisfactory description of DP-Mes, and that the additional modes in the calculation are not coincidental but more likely absent in the experiment due to Raman enhancement and environmental factors.

To further validate the structural sensitivity of our theoretical framework, we performed the same analysis for a trajectory initiated in the $^1TT$ state, generating its intrinsic vibrational structure. In analogy to the coherence transfer spectrum, we find that the dominantly displaced modes belong again to $A_1$ symmetry, with negligible contributions of other modes. The resonance Raman spectrum resulting from this intrinsic calculation (Fig. 4d, orange) again predicts some modes which are not experimentally observed (see above), but importantly reproduces every experimental $^1TT$ frequency. Comparing the relative intensities between the experimental and theoretical spectrum is subject to the same resonance Raman considerations mentioned above. Importantly, the more complex nature of the intrinsic $^1TT$ Raman experiment is expected to result in larger deviations in the relative mode intensities compared with the calculation due to multiple resonance Raman enhancement effects. We point the reader to Supplementary Discussion, section 3, where we present a more in-depth discussion.

Our simulations further allow us to compare directly the effect of initiating the trajectory on $S_1$ or $^1TT$ without the added complication of varying resonance Raman factors. In the high-frequency region (>1000 cm$^{-1}$), we observe a near-perfect intensity match with marginal intensity differences only noticeable for the 1207 cm$^{-1}$ mode. Crucially, in the low-frequency region, the simulations reproduce our surprising observation of a new low-frequency mode at 127 cm$^{-1}$ that only appears following coherence transfer from $S_1$ (Fig. 3), as a signature of singlet fission. Despite any ambiguities regarding optical selection rules or the role of environment, comparison between $S_1$-initiated and $^1TT$-initiated vibrational coherence reveals the same essential results in experiment and theory: singlet fission in DP-Mes enables transfer of vibrational coherence, and this coherence carries unique signatures of the passage of the wavepacket. Thus we consider that our simulations provide an excellent description of the tuning modes in DP-Mes. We further recall that the simulations closely reproduce the electronic dynamics[48], including the mediating role of virtual charge-transfer states[46,47]. As these dynamics depend sensitively on the interplay of tuning and

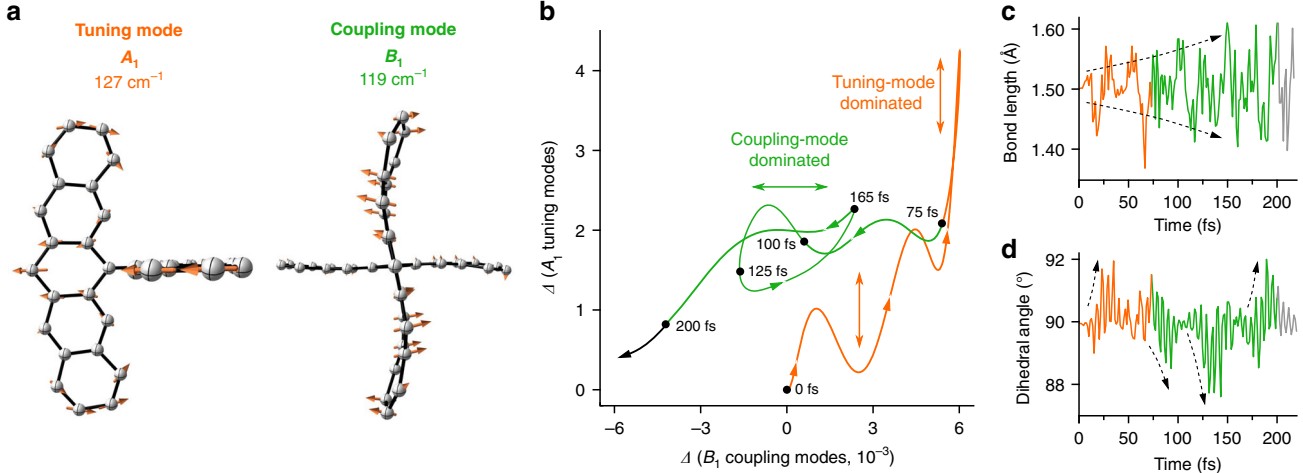

**Fig. 5** Coupling and tuning-mode behaviour in DP-Mes. **a** Representative normal-mode examples of a tuning (left) and a coupling mode (right). The molecular structure was truncated to the core pentacene units for clarity. **b** Correlation plot of the total time-dependent displacement amplitude of $A_1$ tuning and $B_1$ coupling modes. Contour arrows indicate the direction of evolution and black dots indicate the time of the trajectory. The initial motion is dominated by tuning modes (orange vertical arrows) before a rapid damping of the tuning modes funnels the energy into the coupling modes (green horizontal arrow). **c** Central pentacene–pentacene bond length and **d** dihedral angle during the ultrafast singlet fission reaction. Traces are colour coded to match panel (**b**) and dashed arrows indicate significant increases in the values of the parameters

coupling modes, our model description of the coupling modes is at a minimum qualitatively correct.

**Character of coupling and tuning modes**. Motivated by the striking experimental and theoretical structural agreement, we can assign the specific role of each observed mode governing the singlet fission process in DP-Mes. Figure 4b, c emphasise that all experimentally observed modes in the coherence transfer spectrum are of $A_1$ symmetry, and consequently do not mediate the coupling between $S_1$ and $^1TT$. Displacements of these modes alter the energies of the states but the vibronic couplings between them remain zero (see Supplementary Discussion, section 1, and Supplementary Table 3). The experimental coherence transfer spectrum therefore represents the most displaced tuning modes of the singlet fission reaction. From our ab initio simulations, we can immediately show that their atomistic vibrational motions are primarily associated with in-plane ring deformations affecting each pentacene moiety as well as modulating the pentacene–pentacene central bond lengths (Fig. 5a, left). This is consistent with the behaviour expected for tuning modes in conjugated systems where bond-length alterations are caused by the optically induced $\pi \rightarrow \pi^*$ transition to energetically relax the molecule[8,74,75].

According to our TTNS simulations the most active coupling modes of the reaction are instead of $B_1$ and $B_2$ symmetry, albeit with predicted displacements that are at least two orders of magnitude smaller than the tuning modes (Fig. 4b). In line with this result, our experimental resonance Raman spectra reveal no peaks directly attributable to coupling modes. The quadratic dependence of the Raman intensity on the mode displacements (Eq. (1)) implies these would be weaker in intensity by at least 4 orders of magnitude, which is too small to be detected even with our high signal-to-noise ratio. This observation, specifically illustrated here for DP-Mes, likely explains the general lack of experimentally observed coupling modes in singlet fission and similar condensed-phase surface crossing reactions in the literature.

While we cannot experimentally monitor the coupling modes, preventing us from directly validating their simulation parameters, we emphasise that our simulations accurately describe the singlet fission dynamics as well as the transfer of vibrational

coherence (including the unique enhancement of low-frequency modes). These observables are governed by the precise interplay of both tuning and coupling modes, such that, if the coupling mode description in our simulations were inadequate, the model would fail to reproduce the overall dynamics and coherence transfer characteristics. We carried out additional simulations by systematically varying the coupling mode strength, which confirm the notion that these degrees of freedom are strongly linked and that the coupling mode description is well constrained. An in-depth discussion of these results is provided in the Supplementary Discussion, section 1.6, and Supplementary Fig. 8.

A fundamental limitation in theoretically describing a fully structural model of electronic dynamics is that the model contains more parameters than observables. Our approach benchmarks the model against not just the timescale of singlet fission but also the spectrum of $^1TT$ vibrational coherence and its sensitivity to the fission pathway, allowing for a greater degree of confidence in the theoretical description than previously achieved. Despite potential for an improved model description, we are confident to extract the main structural character of the coupling modes from our TTNS simulations. We find that the primary effect of the coupling modes is to create a local twist around the pentacene–pentacene bond and thus a deviation from orthogonality (see Fig. 5a, right). This localised twist causes transient, time-dependent wavefunction overlap of the frontier orbitals of each pentacene monomer. The resulting electronic coupling between the pentacene sub-units thereby promotes the super-exchange reaction from $S_1$ to $^1TT$ via the higher-lying CT states[22,46–48].

**Coordinated interplay of coupling and tuning modes**. To understand the interplay of vibrational tuning and coupling coordinates during and following the initial Franck-Condon relaxation (~200 fs, see Supplementary Fig. 10a), we turn to the detailed molecular movie generated in our simulations (see Supplementary Movie 1). Upon investigating the correlation between the collective time-dependent displacements of the $A_1$ (tuning) and $B_1$ (coupling) modes within the first 200 fs (Fig. 5b) we identify two sequential temporal regimes. From 0 to 75 fs, the dynamics are dominated by tuning modes (vertical displacement, orange), which position the energy levels of DP-Mes for efficient

$S_1 \rightarrow {}^1TT$ crossing. Over the subsequent 125 fs, the activity shifts towards the coupling modes (horizontal displacement, green) which drive the conversion. Importantly, during this coupling-mode dominated regime, the tuning-mode displacements are severely damped to further enhance the reaction yield, highlighting that the singlet fission reaction in DP-Mes is dictated by the synchronised motions along these collective coordinates.

To explore the effect of this collective motion on local coordinates, we extracted from our movie the time evolution of two key structure parameters—the central pentacene dimer bond length and associated dihedral angle. We find that the amplitude modulation of the inter-pentacene bond steadily increases until ~150 fs (Fig. 5c, dashed arrows), with the largest increase occurring during the tuning-mode driven period at ~65 fs (orange). Inspection of the movie shows that the local activation of this central bond is furthermore coupled to several in-plane pentacene ring deformations. As the much weaker coupling modes drive DP-Mes away from the initial orthogonal configuration of the two pentacene units (green), this enhanced bond length activity substantially increases the coupling strength between the pentacene π-systems. In contrast, the local dihedral angle between the two pentacene units displays bursts of activity at well-separated ~50 fs intervals during the first 200 fs (Fig. 5c). We find that positive bursts are present in the tuning-mode driven time window (orange), while negative bursts correlate with the coupling-mode driven regime (green and dashed arrows). Such local behaviour depicts a well-defined evolution on the potential energy surfaces during which the molecule is regularly returned to an energetically favourable region where coupling modes can actively drive the reaction. We note that while these motions are initially activated through the singlet fission process, they do not simply track the electronic kinetics but remain active throughout the simulation window (Supplementary Fig. 10).

## Discussion

Our approach provides clarity about the ultrafast intramolecular singlet fission reaction in DP-Mes with significantly improved structural resolution. We employed structurally sensitive excited-state Raman spectroscopy to uncover the transfer of vibrational wavepackets from $S_1$ to ${}^1TT$, mandating a vibrationally coherent reaction mechanism despite there being no direct coupling between these states. Using this detailed kinetic and structural information as a benchmark for a full quantum dynamics simulation enabled us to assign the observed Raman spectrum to the dominant tuning modes of the process. The compelling match between simulation and experiment allowed us further to infer the crucial coupling modes of the system and record the underlying molecular movie for the singlet fission reaction in a complex molecular system of over 100 atoms. While we have focused exclusively on intramolecular singlet fission in this report, both the experimental and theoretical techniques can be equally well applied to a wide range of ultrafast photochemical processes, from charge transfer[1] and photoisomerisation[8] to polaritonic chemistry[76,77]. Together, these methods set up a powerful tool for describing and understanding reaction dynamics beyond the Born-Oppenheimer approximation, providing detailed insight into the reaction mechanism at conical intersections or avoided crossings in complex materials of practical interest. One immediate result from our analysis is that the coupling modes underpinning singlet fission in DP-Mes are far less intense than the tuning modes. In this and likely most other systems, new experimental approaches are needed to directly observe coupling modes. Alternatively, the TTNS simulations could be systematically improved to explore the coupling mode parameter spaces within the linear vibronic Hamiltonian description—the quality

of which underpins the full simulation. The robustness of the comparison to experiment could be further enhanced by direct incorporation of finite-temperature and environmental effects within the TTNS method, or incorporation of the relevant Franck-Condon factors[73] in the transformation into resonance Raman spectra.

Beyond establishing detection limits for mechanistically relevant vibrational modes, our results report on the functionality of coupled vibrational and electronic dynamics in ultrafast reactions. The photoexcited system evolves along a precisely synchronised set of multiple vibrational modes which must act in concert (i.e. coherently) to promote a highly efficient ultrafast reaction. Taken together with recent results on a vibrational-phase effect on the ultrafast photoisomerisation reaction in rhodopsin[8], our results strongly support the notion that the vibrationally coherent evolution out of the Franck-Condon region after photoexcitation offers an opportunity to tune the outcome of any ultrafast (<10 ps) reaction through rational design[2]. Indeed, the ability to visualise molecular motion with atomistic detail offers prediction scope for systems that exhibit a strong connection between functionality and real-space motion, paving the way for the discovery of novel functional materials. We anticipate the techniques presented above will enable significant advances in the understanding of ultrafast phenomena in general, from charge generation in solar cells[78] to biological light harvesting[79].

## Methods

**Materials**. DP-Mes was synthesised following our original protocol[46]. Briefly, The precursor for DP-Mes was 6,6′-bispentacenequinone. A solution of mesityl magnesium bromide solution in ether (1 M, 1.7 mL) was added to a solution of 6,6′-bispenacenequinone (100 mg, 0.171 mmol) in THF (30 mL) at 0 °C under argon atmosphere. The reaction mixture was brought to room temperature and stirred for 48 h. After quenching with HCl (1 M, 2 mL), the mixture was diluted with ether (60 mL) and washed with water (10 mL). After solvent evaporation, the crude product was purified with column chromatography on silica gel (DCM:hexane, 1:15 v/v) to give target compound DP-Mes as a deep blue solid (99 mg, 73% yield). To prepare pure thin films of DP-Mes, a stock solution of DP-Mes in toluene (10 mg/mL; solvent was purchased from Sigma Aldrich and used as received) was cast on thin microscope coverslips (150 μm, borosilicate glass) and spun at 800 rpm for 60 s. Films were encapsulated within a nitrogen-filled glove-box to enhance long-term stability.

**Impulsive vibrational spectroscopy setup**. The impulsive vibrational spectroscopy setup has been described in detail elsewhere[11,66,80]. Briefly, a Yb:KGW amplifier system (Light Conversion, Pharos, 5 W, 10 kHz) provides pulses centred at 1030 nm with a pulse duration of ~200 fs. A small portion is used to generate a chirped white light continuum in a 3 mm sapphire window used as the probe pulse in all experiments, with a Gaussian beam diameter at full-width-half-maximum (fwhm) of 30 μm at the sample. Impulsive pump pulses in the near-IR (150 nJ, 50 μm fwhm) and visible (130 nJ, 70 μm fwhm) were generated by previously reported home-built non-collinear parametric amplifiers (NOPAs)[81]. To produce the narrow-band pump pulse (80 nJ, 70 μm fwhm) used for the preparation of the triplet state (Fig. 3) we temporally stretched the seed white light continuum in the visible pump NOPA with a BK7 rod (7 cm length) prior to amplification resulting in ~200 fs pulses. The duration is limited by the employed pump pulses (third harmonic) which was derived directly from the amplifier system.

**Non-resonant impulsive vibrational spectroscopy**. To gain access to the ground-state Raman spectrum, we employed a temporally compressed 12 fs pump pulse tuned to 800 nm, which generates vibrational coherences via impulsive stimulated Raman scattering exclusively on the ground electronic state due to the lack of an electronic resonance (see Fig. 3c, right). Fourier transformation of the detected coherent oscillations over the absorption spectrum of DP-Mes allows us to independently measure a time-domain Raman spectrum which is directly comparable to resonant impulsive Raman spectra when probed in the same wavelength region[64,66,68,82].

**Excited-state impulsive vibrational spectroscopy**. Intrinsic impulsive Raman reference measurements on ${}^1TT$ were carried out by first photoexciting the sample (600 nm, 200 fs) to generate a population in $S_1$ that was allowed to undergo singlet fission. After a time delay of 9 ps, at which point the singlet fission reaction is complete[47], the ${}^1TT$ population was re-excited with an impulsive Raman pump pulse (800 nm, 11 fs) resonant with the excited-state absorption observed >750 nm

(see also Supplementary Fig. 9a)[47]. The impulsively generated vibrational coherence was subsequently recorded in the excited-state absorption region (515–525 nm). Following subtraction of electronic kinetics and the background ground-state vibrational activity, a Fourier transform provided the intrinsic triplet Raman signature (Fig. 3, purple). We remark that no significant vibrational activity is detected in the ground-state bleach or stimulated emission regions after the subtraction procedure.

**Vibrational spectroscopy data analysis.** The experimental impulsive vibrational spectroscopy data sets were processed according to previously published procedures[66]. Briefly, after chirp-correction, all traces were truncated in time to include only positive time delays >210 fs to prevent coherent artefact contributions affecting the signal. We subsequently extracted the residual oscillations for each probe wavelength by globally fitting the experimentally recorded maps to a sum of two exponentially decaying functions with an offset. The coherent oscillations were further truncated to an overall time length of 1.28 ps prior to apodization (Kaiser-Bessel window, $\beta = 1$), zero-padding (3×) and Fourier transformation. The frequency resolution corresponded to ~26 cm$^{-1}$ and the lowest resolvable frequency was ~52 cm$^{-1}$. To extract the intrinsic $^1$TT Raman spectrum, we recorded the vibrational coherence in the presence and absence of the actinic pump pulse and subsequently subtracted the two coherences in the time-domain to minimize ground-state contributions, as outlined previously. The remaining data analysis was the same. Care was taken to ensure that all traces were temporally aligned using reference measurements on toluene to exclude possible Fourier artefacts in the comparison between different experiments.

**Computational methods.** Concerning the parametrisation of the model and the methodology for the time evolution, the following is very closely based on Schröder et al.[48] and the supplementary information of that work, and is presented here for the reader's convenience. The interested reader should refer to that work for further detail on the methods used. For the time evolution of the system we employ a linear vibronic model, which makes use of the harmonic approximation. This includes two purely electronic and vibrational terms, as well as an electron-phonon coupling term:

$$H = H_{\text{el}} + \sum_{n=1}^{256} W_n \frac{\hat{b}_n^\dagger + \hat{b}_n}{\sqrt{2}} + \omega_n \hat{b}_n^\dagger \hat{b}_n. \quad (2)$$

Here $\hat{b}_n^\dagger$, $\hat{b}_n$ are the creation and annihilation operators of vibrational mode $n$, with $\omega_n$ its frequency, and $W_n$ a matrix describing its effect on the electronic energies/couplings upon displacement.

The Hamiltonian is written in the basis of the five electronic states:

$$\{|TT\rangle, |LE_+\rangle, |LE_-\rangle, |CT_+\rangle, |CT_-\rangle\}. \quad (3)$$

The above locally excited (LE) and charge transfer (CT) states are symmetry-pure, and are written as linear combinations of the LE states of each pentacene unit, as well as CT states corresponding to electron transfer from one monomer to the other:

$$|LE_\pm\rangle = \frac{1}{\sqrt{2}}(|LE_A\rangle \pm |LE_B\rangle) \quad (4)$$

$$|CT_\pm\rangle = \frac{1}{\sqrt{2}}(|CT_A\rangle \pm |CT_B\rangle) \quad (5)$$

The states corresponding to the two monomers may be written as Slater determinants of the monomer-localised HOMOs/LUMOs: $h_A$, $h_B$, $l_A$, $l_B$ shown in Supplementary Fig. 1. In particular:

$$|LE_A\rangle = \frac{1}{\sqrt{2}}(|h_A\alpha l_A\beta h_B\alpha h_B\beta| - |h_A\beta l_A\alpha h_B\alpha h_B\beta|) \quad (6)$$

$$|LE_B\rangle = \frac{1}{\sqrt{2}}(|h_A\alpha h_A\beta h_B\alpha l_B\beta| - |h_A\alpha h_A\beta h_B\beta l_B\alpha|) \quad (7)$$

$$|CT_A\rangle = \frac{1}{\sqrt{2}}(|h_A\alpha l_B\beta h_B\alpha h_B\beta| - |h_A\beta l_B\alpha h_B\alpha h_B\beta|) \quad (8)$$

$$|CT_B\rangle = \frac{1}{\sqrt{2}}(|h_A\alpha h_A\beta h_B\alpha l_A\beta| - |h_A\alpha h_A\beta h_B\beta l_A\alpha|) \quad (9)$$

The correlated triplet pair state may be written as:

$$|TT\rangle = \frac{1}{\sqrt{3}}[|h_A\alpha l_A\alpha h_B\beta l_B\beta| + |h_A\beta l_A\beta h_B\alpha l_B\alpha| \\ - \frac{1}{2}(|h_A\alpha l_A\beta h_B\alpha l_B\beta| + |h_A\alpha l_A\beta h_B\beta l_B\alpha| \\ + |h_A\beta l_A\alpha h_B\alpha l_B\beta| + |h_A\beta l_A\alpha h_B\beta l_B\alpha|)] \quad (10)$$

Since DP-Mes has a $D_{2d}$ symmetry, we can classify the electronic states according to irreducible representations of the point group, by operating on them with the various symmetry operators which are summarised in Supplementary Fig. 1. The energies of $|LE_\pm\rangle$, $|CT_\pm\rangle$ are calculated within TD-DFT, using the range-separated functional LC-BLYP with an optimised range-separation parameter $\gamma = 0.29$. The energy of the TT state is approximated by that of the spin-two ground state of DP-Mes as visualised in Supplementary Fig. 2. The energies

and symmetries of the electronic states considered here are summarised in Supplementary Table 1.

We included 252 out of the 318 vibrations of DP-Mes and excluded the very slow modes with frequencies below 110 cm$^{-1}$ as these are highly anharmonic. Likewise, we excluded the very high-frequency C–H stretches above 1680 cm$^{-1}$ as they are expected to make little contribution to the overall dynamics and they are beyond the scope of our experimental technique. The molecular vibrations were calculated at the cc-pVDZ, B3LYP level of DFT, using the NWChem software[83]. We refer the reader to the Supplementary Discussion, section 1, for further information on the transformation of the vibrational environments, Tree Tensor Network States, real-time displacement of molecular vibrations, conversion of displacements into resonance Raman spectra, a discussion on the importance of a large vibrational bath and a coupling mode assignment.

## Data availability
The data underlying all figures in the main text and supplementary information are publicly available at https://doi.org/10.17863/CAM.42998.

## Code availability
The custom tree tensor networks state code used for simulations and the custom code for computing real-time displacements of physical normal modes are available by written request to alex.chin@insp.jussieu.fr.

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

## Acknowledgements

This work was supported by the Engineering and Physical Sciences Research Council, UK (Grant Numbers EP/M025330/1, EP/M01083X/1, EP/L015552/1 and EP/M006360/1) and the Winton Programme for the Physics of Sustainability. J.W. acknowledges financial support from Singapore MOE Tier 3 Programme (MOE2014-T3-1-004). T.W. acknowledges the Marie Curie Intra European Fellowship (PIEF-GA-2013-623652) within the 7th European Community Framework Programme. C.S. acknowledges financial support by the Royal Commission for the Exhibition of 1851.

## Author contributions

A.J.M. conceived the study and, together with C.S., designed the experiments. C.S. carried out time-resolved measurements, assisted by T.W. DP-Mes was synthesised by J.F. and J.W., and S.L. prepared samples. A.J.M., C.S., T.W. and S.L. analysed the data. TTNS model was developed by F.A.Y.N.S., D.H.P.T., N.D.M.H. and A.W.C. and refined by A.M.A. All authors interpreted the results. A.J.M., C.S. and A.M.A. wrote the paper.

## Additional information

**Competing interests:** The authors declare no competing interests.

