## [Peer Review File · Nature Communications]

Reviewers' comments:

Reviewer #2 (Remarks to the Author):

The revised manuscript (originally submitted to Nat. Chem.) has been thoroughly revised. It seems to me that authors have carefully addressed most of my concerns and those by the other two reviewers. This combined experimental/theoretical work will be of value to the singlet fission research field. I support publication as in in Nat. Commun.

Reviewer #3 (Remarks to the Author):

The revised submission in combination with the response by the authors has addressed some of the points raised in my previous report, but still leaves me with a number of critical concerns.

The authors have clarified that the merit of the submitted work lies not in providing new insights into singlet fission, but in presenting a combination of time- and frequency-domain Raman spectroscopy on one hand, and TTNS simulations on the other. In their response, the authors call this combination a "new framework for understanding ultrafast non-adiabatic processes" and "a fundamental advance". I find these statements rather bold, as I would normally reserve such language for introducing new concepts upon which to understand certain phenomena (such as the conception of BCS theory for understanding superconductivity; pardon the hyperbolic example). The submitted work introduces a new combination of techniques (which have individually been applied before), but at the fundamental level I would call this common practice in science - people come up with new combinations of techniques all the time. I do not mean to discredit the high level of experimental and theoretical work found in this submission, nor do I wish to start a semantic discussion, but I do not agree with the fundamental degree of novelty that these statements seem to predicate.

With regard to my previous point 1, there may have been a misunderstanding in that it was never the experimental peak at 127 cm^{-1} that I questioned, but the calculated peak. The authors claim

that all experimental peaks are reproduced by the calculations, including the one at 127 cm⁻¹. My source of concern is that the calculated spectrum features many spurious peaks that are not observed in the measurements, and which are not accounted for. How do we know that the calculated peak at 127 cm⁻¹ is not due to a similar artifact? This is an important question to ask, since if it is spurious as well, the calculations completely fail in reproducing any of the spectral differences between the intrinsic and the transferred spectrum. And of course, the larger the number of spurious peaks, the larger the probability for one to conveniently show up at a certain location. I am not asking the authors to verify the validity of this peak beyond the shadow of a doubt (which might well be impossible), but think that such uncertainties should not be shoved under the carpet in their discussions.

On a related note: how quantitative are the calculations in reproducing the experimentally observed reorganization energies? The spectra in Fig. 4 are all in arbitrary units, making this hard to assess.

Reviewer 2 has expressed similar concerns regarding the quantitative accuracy of the theoretically calculated coupling modes, which the authors have addressed by showing a remarkable sensitivity of the singlet fission dynamics to the strength of these modes. The fact that the simulations are found to reproduce the experimentally observed fission rate was then used as an argument to proof that the calculations are at least semi-quantitative. I find this rationale a bit of a leap, since one uses a single observable to benchmark many dozens of coupling modes. Another way of looking at this situation is that the quantitative accuracy in parameterizing these modes is all the more important, since small fluctuations can give arbitrary different outcomes for the fission dynamics. Again, the question arises to what extent the results are coincidental. And again, I do not ask the authors to give definite proof of their claims, but instead pay more attention to these uncertainties in their discussions.

Reviewer #2 (Remarks to the Author):

The revised manuscript (originally submitted to Nat. Chem.) has been thoroughly revised. It seems to me that authors have carefully addressed most of my concerns and those by the other two reviewers. This combined experimental/theoretical work will be of value to the singlet fission research field. I support publication as in in Nat. Commun.

We thank the reviewer for the positive evaluation of our manuscript and are glad that we could address all of the concerns satisfactorily.

Reviewer #3 (Remarks to the Author):

The revised submission in combination with the response by the authors has addressed some of the points raised in my previous report, but still leaves me with a number of critical concerns.

The authors have clarified that the merit of the submitted work lies not in providing new insights into singlet fission, but in presenting a combination of time- and frequency-domain Raman spectroscopy on one hand, and TTNS simulations on the other. In their response, the authors call this combination a "new framework for understanding ultrafast non-adiabatic processes" and "a fundamental advance". I find these statements rather bold, as I would normally reserve such language for introducing new concepts upon which to understand certain phenomena (such as the conception of BCS theory for understanding superconductivity; pardon the hyperbolic example). The submitted work introduces a new combination of techniques (which have individually been applied before), but at the fundamental level I would call this common practice in science - people come up with new combinations of techniques all the time. I do not mean to discredit the high level of experimental and theoretical work found in this submission, nor do I wish to start a semantic discussion, but I do not agree with the fundamental degree of novelty that these statements seem to predicate.

We appreciate the reviewer's concern regarding the presentation of our work. It is true that applying and combining new techniques is a common scientific practice, but we maintain that the specific combination we have applied here allows an advance that is, in our view, beyond the ordinary. However, we recognise that such determinations are subjective and share the reviewer's desire to avoid a semantic debate.

We have thus moderated our language regarding the fundamental novelty of our work throughout the manuscript.

With regard to my previous point 1, there may have been a misunderstanding in that it was never the experimental peak at 127 cm^{-1} that I questioned, but the calculated peak. The authors claim that all experimental peaks are reproduced by the calculations, including the one at 127 cm^{-1} . My source of concern is that the calculated spectrum features many spurious peaks that are not observed in the measurements, and which are not accounted for. How do we know that the calculated peak at 127 cm^{-1} is not due to a similar artifact? This is an important question to ask, since if it is spurious as well, the calculations completely fail in reproducing any of the spectral differences between the intrinsic and the transferred spectrum. And of course, the larger the number of spurious peaks, the larger the probability for one to conveniently show up at a certain location. I am not asking the authors to verify the validity of this peak beyond the shadow of a doubt (which might well be impossible), but think that such uncertainties should not be shoved under the carpet in their discussions.

We thank the reviewer for the clarification of this point; we had indeed misunderstood the earlier remark. It is our view that none of the peaks present in the calculated spectrum are spurious or related to artefacts. They all correspond to real motions of the molecule, and despite their low approximated Raman intensity these modes are highly displaced. Specifically, the mode at 127 cm^{-1} is the most absolute displaced of all 252 modes considered in our model (but exhibits a smaller resonance Raman intensity due to the frequency scaling in Equation 1). This is shown directly in Figure R1 below, which is now included in the Supplementary Information.

Figure R1| Mode displacements extracted from TTNS simulation. Displacement amplitude Δ is obtained from the Fourier transform of the calculated displacements of transferred vibrational coherence over a 1 ps window (kinetic traces in main-text Figure 4b).

A key point in support of these modes (including at 127 cm^{-1}) being 'real' is that they reproduce one of our key experimental observations: the vibrational coherence

detected in the TT state is sensitive to whether it is 'intrinsic' or 'transferred'. If the low-frequency modes were simply an artefact of our TTNS approach or our parameterisation of the molecule, we would expect them to appear in both simulation types. Yet, the simulation reproduces the general behaviour observed in the experiment, namely the lack of low-frequency modes in the intrinsic Raman spectrum compared to the transferred Raman spectrum

However, we cannot definitively show that these modes are 'real' for the reason already mentioned in our manuscript: the experiment relies on resonance Raman enhancement factors to be able to detect these vibrations in the transient absorption experiment, and not all modes will benefit equally from this enhancement effect.

Possible refinements to our approach to reduce the uncertainty here include: 1) Performing the impulsive vibrational spectroscopy in different spectral regions where other TT transitions may present different Raman enhancement factors (challenging in DP-Mes since no other strong transitions have been reported in the 500-950 nm range readily available in our experiment), or 2) Incorporating resonance enhancement factors into the model to convert displacements into effective Raman spectra. The latter approach is far from trivial and requires not only knowledge of the 5 potential energy surfaces we already included in our calculations, but further higher-excited states associated with the photoinduced absorption we probe. This is at present not feasible within our framework but is certainly an interesting avenue to achieve a quantitative agreement between theory and experiment.

We have incorporated these arguments into the main text at several places to acknowledge the potential discrepancies between the experiment and the theory in light of the reviewer's comment.

On a related note: how quantitative are the calculations in reproducing the experimentally observed reorganization energies? The spectra in Fig. 4 are all in arbitrary units, making this hard to assess.

We are unsure to what the reviewer is referring. In Figure 4 we show Raman intensity spectra and not reorganisation energies. While we have not extracted reorganisation energies from these results, the experimental and theoretical spectra can be directly compared (both are processed and normalised following similar procedures and are presented on common frequency axes) and the close match indicates that the tuning modes are well reproduced in our model.

Reviewer 2 has expressed similar concerns regarding the quantitative accuracy of the theoretically calculated coupling modes, which the authors have addressed by showing a remarkable sensitivity of the singlet fission dynamics to the strength of these modes. The fact that the simulations are found to reproduce the experimentally observed fission rate

was then used as an argument to proof that the calculations are at least semi-quantitative. I find this rationale a bit of a leap, since one uses a single observable to benchmark many dozens of coupling modes. Another way of looking at this situation is that the quantitative accuracy in parameterizing these modes is all the more important, since small fluctuations can give arbitrary different outcomes for the fission dynamics. Again, the question arises to what extent the results are coincidental. And again, I do not ask the authors to give definite proof of their claims, but instead pay more attention to these uncertainties in their discussions.

The reviewer's general point is a fair one, and indeed this is a major rationale for our study. It is common for high-level simulations of ultrafast processes to include many parameters for electron-phonon couplings, couplings between electronic states, etc., with the quality of the description often judged solely on its match to a single experimental electronic population kinetic. This is precisely the approach the reviewer describes, using a single observable to benchmark many dozens (or more) of parameters or smaller numbers of 'proxy' parameters with no direct physical translation. What we have sought to do in our approach is provide further benchmarking of the molecular parameters, through additional stringent restrictions such as the match of tuning modes, coherence transfer and the difference between 'intrinsic' and 'transferred' coherence. This provides greater confidence in the accuracy of the model, the quality of which relies not on the TTNS simulations but the underlying linear vibronic Hamiltonian. As the reviewer notes, despite this improved confidence we cannot confirm with certainty that our description is not coincidental. To do so would require either direct observation of particular coupling modes (which our calculations suggest is not experimentally feasible in DP-Mes, and would anyway simply shift the uncertainty to those modes which are still not observed) or calculations of all combinations of coupling mode parameters to ascertain the likelihood that the good agreement we observe is not coincidental.

We have incorporated a brief discussion of these considerations into the main text, both to highlight the current limitations of the approach and how our methods might be built upon to address them in the hope to give credit to the reviewer's comment.

REVIEWERS' COMMENTS:

Reviewer #3 (Remarks to the Author):

I am satisfied with the way the authors have addressed the concerns raised in my previous reports. I support publication of this work in Nature Communications.